EMBO
reports

# PP2A-B56 binds to Apc1 and promotes Cdc20 association with the APC/C ubiquitin ligase in mitosis

Kazuyuki Fujimitsu & Hiroyuki Yamano*

## Abstract

Cell cycle progression and genome stability are regulated by a ubiquitin ligase, the anaphase-promoting complex/cyclosome (APC/C). Cyclin-dependent kinase 1 (Cdk1) has long been implicated in APC/C activation; however, the molecular mechanisms of governing this process *in vivo* are largely unknown. Recently, a Cdk1-dependent phosphorylation relay within Apc3-Apc1 subunits has been shown to alleviate Apc1-mediated auto-inhibition by which a mitotic APC/C co-activator Cdc20 binds to and activates the APC/C. However, the underlying mechanism for dephosphorylation of Cdc20 and APC/C remains elusive. Here, we show that a disordered loop domain of Apc1 (Apc1-loop$^{500}$) directly binds the B56 regulatory subunit of protein phosphatase 2A (PP2A) and stimulates Cdc20 loading to the APC/C. Using the APC/C reconstitution system in *Xenopus* egg extracts, we demonstrate that mutations in Apc1-loop$^{500}$ that abolish B56 binding decrease Cdc20 loading and APC/C-dependent ubiquitylation. Conversely, a non-phosphorylatable mutant Cdc20 can efficiently bind the APC/C even when PP2A-B56 binding is impeded. Furthermore, PP2A-B56 preferentially dephosphorylates Cdc20 over the Apc1 inhibitory domain. These results indicate that Apc1-loop$^{500}$ plays a role in dephosphorylating Cdc20, promoting APC/C-Cdc20 complex formation in mitosis.

Keywords APC/C; Cdc20; Cdk; cell cycle; PP2A
Subject Category Cell Cycle

## Introduction

The cell cycle is a biological process by which genetic information is accurately duplicated and segregated into daughter cells. At the same time, growth, proliferation and differentiation are controlled accordingly in response to environmental cues, ensuring genome stability. Cyclin-dependent kinases (Cdks) and the APC/C are some of the most significant cell cycle regulators, controlling the cell cycle through phosphorylation and ubiquitylation, respectively [1–4]. Cdk activity is vital for DNA replication and driving mitosis, while the APC/C-dependent ubiquitylation and proteolysis of several key regulatory proteins at specific times are essential for mitotic progression. For example, APC/C-dependent proteolysis of securin and cyclin B drives sister chromatid separation and mitotic exit. The APC/C also controls non-mitotic events such as gene expression and differentiation, and even functions in non-dividing cells, such as neurons [5–8].

Cdk1 and APC/C mutually regulate each other in mitosis, forming a negative feedback loop by which a periodicity of cell division cycle is ensured; there can be no mitotic APC/C activation without Cdk1, and there can be no cyclin destruction and no cell division without APC/C. However, despite the obvious importance of the whole mechanism, the control of APC/C activation and regulation still remains poorly understood. One crucial aspect of APC/C regulation is the engagement of its co-activators Cdc20 and Cdh1 as apo-APC/C by itself is not active. Co-activator engagement is regulated by phosphorylation of both the APC/C and co-activators. Mitotic co-activator Cdc20 preferentially binds and activates hyperphosphorylated APC/C promoting mitotic progression, whereas Cdh1, which is inhibited by phosphorylation, functions in telophase and G1 when Cdk activity is low after cyclin B destruction. High-resolution cryo-EM has uncovered the detailed structure of the APC/C complex (comprising 14 subunits in vertebrates) providing mechanistic insights [9–11]. Furthermore, recent reports revealed more dynamic regulation through the disordered loop domains of the Apc3 and Apc1 subunits (Apc3-loop, 182–451; Apc1-loop$^{300}$, 294–399 in *Xenopus*). Cdk1-driven phosphorylation occurs in a coordinated manner between Apc3 and Apc1. Cdk1-dependent Apc3-loop phosphorylation acts as a scaffold and recruits a Cdk regulatory subunit p9/Cks protein via its anion binding domain. In turn, this engages a ternary complex Cks-Cdk1-cyclin and phosphorylates an auto-inhibitory domain in a distant subunit, Apc1, in an intramolecular phosphorylation relay [12]. The auto-inhibitory domain, Apc1-loop$^{300}$, acts as a molecular switch, and in unphosphorylated APC/C, it interacts with the C-box (a short and conserved motif present in the Cdc20/Fizzy family of co-activators) binding site and sterically hinders engagement of Cdc20. Phosphorylation of Apc1-loop$^{300}$ induces a conformational change, resulting in dissociation of Apc1-loop$^{300}$ from the C-box binding site enabling Cdc20 loading for ubiquitylation catalysis [12–14].

Cell Cycle Control Group, UCL Cancer Institute, University College London, London, UK
*Corresponding author. Tel: +44 020 7679 6498; E-mail: h.yamano@ucl.ac.uk

Phosphorylation of Cdc20 is also an important element that controls the formation of active APC/C-Cdc20 complex. Cdk1-dependent phosphorylation of three N-terminal threonine residues near the C-box (T64, T68 and T79 in *Xenopus*) inhibits Cdc20 loading to the APC/C, presumably by precluding Cdc20-Apc6 interaction [15]. Notably, in anaphase extracts induced by non-degradable cyclin B, it has been shown that most of the Cdc20 bound to the APC/C evades phosphorylation at T79 even when the activity of Cdk1 is high and the APC/C is highly phosphorylated. However, exactly how Cdc20 is dephosphorylated in order to bind the APC/C remains elusive although PP2A has been shown to be involved [15].

PP2A is a highly conserved serine/threonine phosphatase consisting of three subunits: catalytic C, scaffolding A and a variable regulatory B subunit [16]. Substrate specificity and/or subcellular localisation are regulated by its associated B subunits. The B subunit family comprises four distinct subfamilies (B/B55, B′/B56, B″/PR72 and B‴/STRN) with multiple isoforms for each subfamily. It is well established that the activity of PP2A-B55δ is vital for mitotic exit [17]. Crucially, the activity of PP2A-B55δ fluctuates during the cell cycle, via the PP2A-B55δ/ENSA/Greatwall pathway, and is high in interphase and low in mitosis [18–20]. Thus, PP2A-B55δ is unlikely to be the phosphatase that initiates Cdc20 dephosphorylation in mid-mitosis although PP2A-B55δ has been reported to be important for Cdc20 dephosphorylation at mitotic exit once it is activated upon Cdk1 inactivation [18–20]. In contrast, PP2A-B56 has been shown to be active during mitosis. In fact, several mitotic events such as silencing of spindle assembly checkpoint (SAC), loading of an E2 ubiquitin-conjugating enzyme to APC/C and assembly of the central spindle are facilitated by PP2A-B56 [21–24]. More recent work suggests that PP2A-B56 achieves its functional specificity by binding to a LxxIxE short linear motif (SLiM) in which Glu (E) at position 6 is crucial, but position 1 (Leu) and position 4 (Ile) can be replaced by other hydrophobic residues, and phosphorylation or the enrichment of acidic residues within and downstream from the motif can increase PP2A-B56 binding [25–27].

We have identified a B56 binding site within the *Xenopus* APC/C complex, which is located in another flexible disordered loop domain of Apc1 (Apc1-loop[500]). Using egg extract of *Xenopus laevis* and reconstitution of apo-APC/Cs in the MultiBac system, we show here that Apc1-loop[500] has a role in PP2A-B56 recruitment in mitosis, which in turn dephosphorylates Cdc20 and controls its loading for APC/C activation. Consistently, phosphorylation site mutant Cdc20 can bind sufficiently to the APC/C independently of PP2A-B56 binding. Furthermore, PP2A-B56 dephosphorylates Cdc20 more efficiently than the Apc1-loop[300]. Our work reveals a mechanism explaining how a mitotic co-activator Cdc20 can specifically bind to and activate the APC/C in anaphase and therefore initiate sister chromatid separation and mitotic exit.

## Results and Discussion

### PP2A B56 regulatory subunit binds to Apc1-loop[500]

Although it has been demonstrated that PP2A is involved in APC/C regulation [15,28,29], the underlying mechanisms have not been well characterised. Structural studies of the APC/C hinted that there are putative disordered loop domains in the APC/C complex

in addition to Apc3-loop and Apc1-loop[300]. We therefore hypothesised that these flexible disordered loop domains could also regulate APC/C activity. It has been recently reported that PP2A-B56 recognises and binds a LxxIxE SLiM on target substrates [25–27]. This finding prompted us to investigate whether a B56 binding site is present in APC/C subunits, in particular, within these disordered loop domains. Primary sequence inspection of those domains has identified one such SLiM (LSPVPE) in a predicted loop domain of Apc1 (515–584 in *Xenopus* Apc1, hereinafter referred to as Apc1-loop[500]) that is located in the N-terminal WD40 domain of Apc1. This sequence is highly conserved among species including human Apc1 (Fig 1A). To verify the ability of this loop domain to bind B56 subunit, we prepared maltose binding protein (MBP) fused to Apc1-loop[500] and its derivatives with mutations such as an 11 residue deletion of the B56 binding site (Δ11) or substitution of three alanines of putative Cdk sites (3A) (Fig 1B) and examined the ability to bind B56γ, a subtype of B56, using *Xenopus* egg extracts (Fig 1C). Pull-down assays showed only wild-type (WT) Apc1-loop[500] significantly bound [35]S-labelled *in vitro*-translated B56γ in anaphase extracts (lane 14), compared with MBP-alone or MBP-fused Apc1-loop[500-Δ11] (lanes 13 and 15). This result suggests that this flexible Apc1-loop[500] domain can bind B56γ via the B56 binding site. This interaction was not observed in interphase extracts, which have low Cdk activity (lane 10), suggesting that the binding of B56 to Apc1-loop[500] depends on phosphorylation of the loop domain. Consistently, non-phosphorylatable Apc1-loop[500-3A] failed to bind B56γ even in anaphase extract (lane 16). We also performed Cdk-dependent *in vitro* kinase assay and confirmed that WT Apc1-loop[500], but not Apc1-loop[500-3A], was efficiently phosphorylated by Cdk2-cyclin A (Fig 1D). Furthermore, we have investigated whether Cdk phosphorylation of Apc1-loop[500] promotes B56 loading. Purified MBP-fused WT Apc1-loop[500], but not 3A, increased its binding affinity to B56, depending on Cdk phosphorylation (Fig 1E and F). We also made Apc1-loop[500] with S558A single alanine substitution of Cdk site within the B56 binding motif (Fig EV1A). Pull-down assays showed that the point mutant S558A abolished B56 binding as efficiently as the 3A mutations (Fig EV1B). This is consistent with the previous report that phosphorylation of the SP site in the middle of the B56 binding site increases binding strength [29]. To further investigate B56 and Apc1-loop[500] interaction, we generated another Apc1-loop[500] mutant protein that harbours two alanine substitutions within the B56 binding site in Apc1-loop[500] (Apc1-loop[500-L557A/V560A]). As was seen for Apc1-loop[500-Δ11], the mutations in the B56 binding site (Apc1-loop[500 L557A/V560A]) abolished the ability to bind B56γ (Fig EV1C, lanes 14–16). As the regulatory B subunit family comprises four distinct subfamilies, B55, B′/B56, B″/PR70 and B‴/STRN, we wanted to examine subfamily specificity. Apc1-loop[500] specifically bound to B56γ, not B55δ or PR70 (Fig EV1C, lanes 13–24). All together, these results demonstrate that upon Cdk phosphorylation, the disordered loop domain Apc1-loop[500] specifically interacts with PP2A harbouring the B56 regulatory subunit.

### Apc1-loop[500] regulates APC/C activity through Cdc20 loading in anaphase

Next, we asked how the B56 binding to Apc1-loop[500] regulates APC/C activity. We have established a functional biochemistry

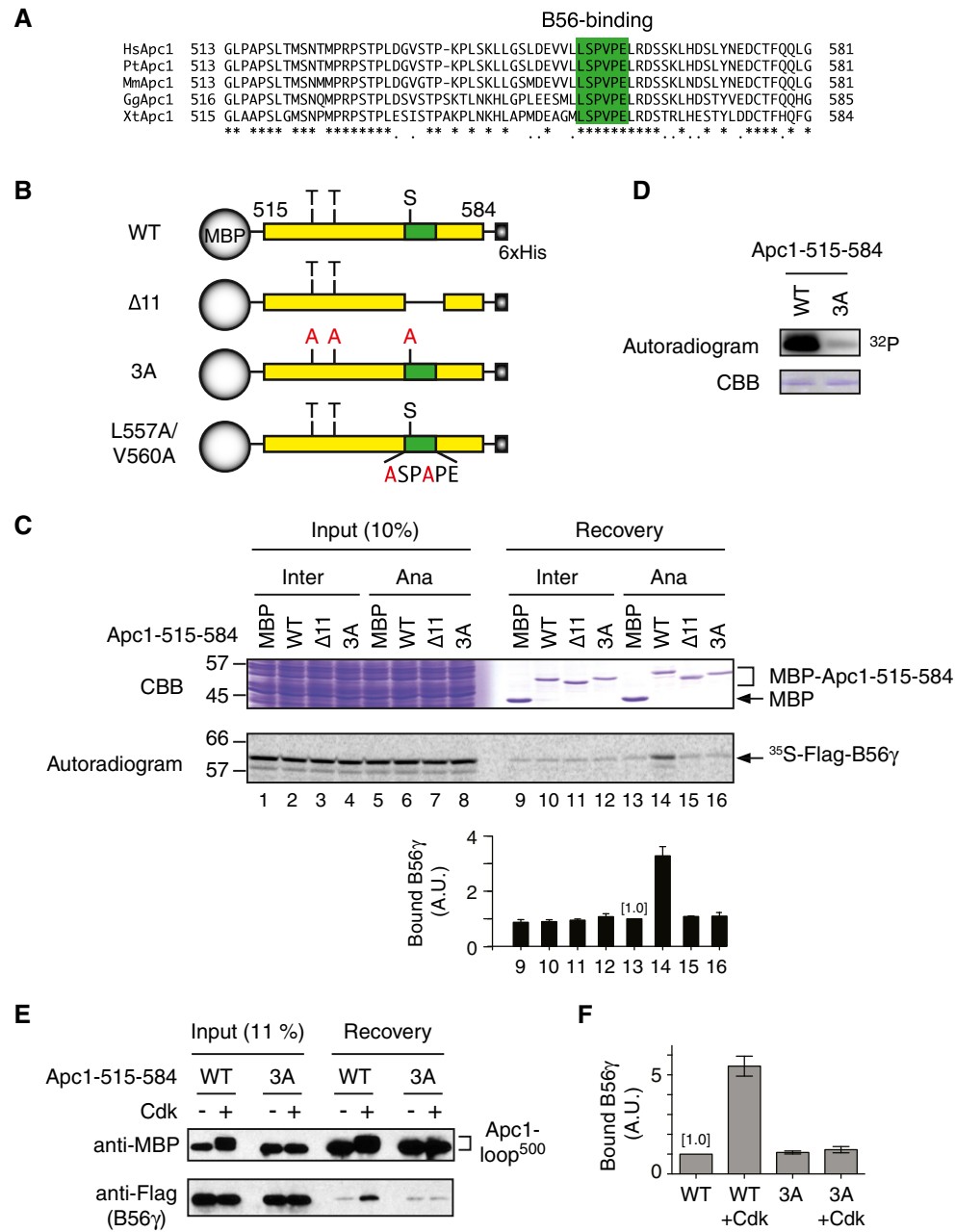

**Figure 1. Cdk-dependent phosphorylation of Apc1-loop$^{500}$ ensures PP2A-B56γ recruitment.**

A  Multiple alignment of a predicted Apc1-loop$^{500}$ domain of vertebrate APC/Cs. The sequences corresponding to residues 515–584 in *Xenopus tropicalis* Apc1 are shown. A putative B56 binding region is coloured in green. Hs, *Homo sapiens* human; Pt, *Pan troglodytes* chimpanzee; Mm, *Mus musculus* mouse; Gg, *Gallus gallus* chicken; Xt, *Xenopus tropicalis* frog.

B  Schematic diagrams of Apc1-loop$^{500}$ constructs. Apc1-loop$^{500}$ (residues 515–584) was fused with maltose binding protein (MBP) at N-terminus and 6xHis at C-terminus. Conserved Cdk phosphorylation sites (SP/TP) are shown as S or T, respectively. The putative B56 binding region is shown in green. The 11-residue [LSPVPELRDST] deletion (Δ11) and alanine substitution mutations to three Cdk phosphorylation sites (3A) or to B56 binding motif (L557A/V560A) are shown.

C  Binding assay using MBP-fused Apc1-loop$^{500}$ fragments and B56γ. Apc1-loop$^{500}$ WT or its derivatives (Δ11 or 3A) were incubated with the $^{35}$S-labelled Flag-B56γ in interphase extract (Inter) or anaphase extract (Ana) supplemented with CycBΔ167 at 23°C for 1 h. The bound proteins were recovered by amylose beads, separated by SDS–PAGE and detected by autoradiography or Coomassie brilliant blue (CBB) staining. The bar graph is quantification of bound B56γ. The intensities of MBP control were arbitrarily set to 1.0. Error bars, SEM from three independent experiments.

D  Cdk-dependent *in vitro* kinase assay of Apc1-loop$^{500}$. MBP-fused WT or 3A Apc1-loop$^{500}$ fragment was incubated with Cdk2-cyclin A in the presence of [γ-$^{32}$P]-ATP at 23°C for 10 min, separated by SDS–PAGE and detected by autoradiography.

E  Cdk can promote Apc1-loop$^{500}$ and B56 interaction. MBP-fused WT or 3A Apc1-loop$^{500}$ fragment was incubated in the presence or absence of Cdk2-cyclin A at 30°C for 60 min. MBP-fused peptides (−/+ kinase) were isolated and incubated with purified Flag-B56γ at 23°C for 30 min. The bound proteins were recovered by amylose beads and analysed by SDS–PAGE and immunoblotting with indicated antibodies.

F  Quantification of (E) Bound B56γ to WT Apc1-loop$^{500}$ control (−Cdk) was arbitrarily set to 1.0. Error bars, SEM from three independent experiments.

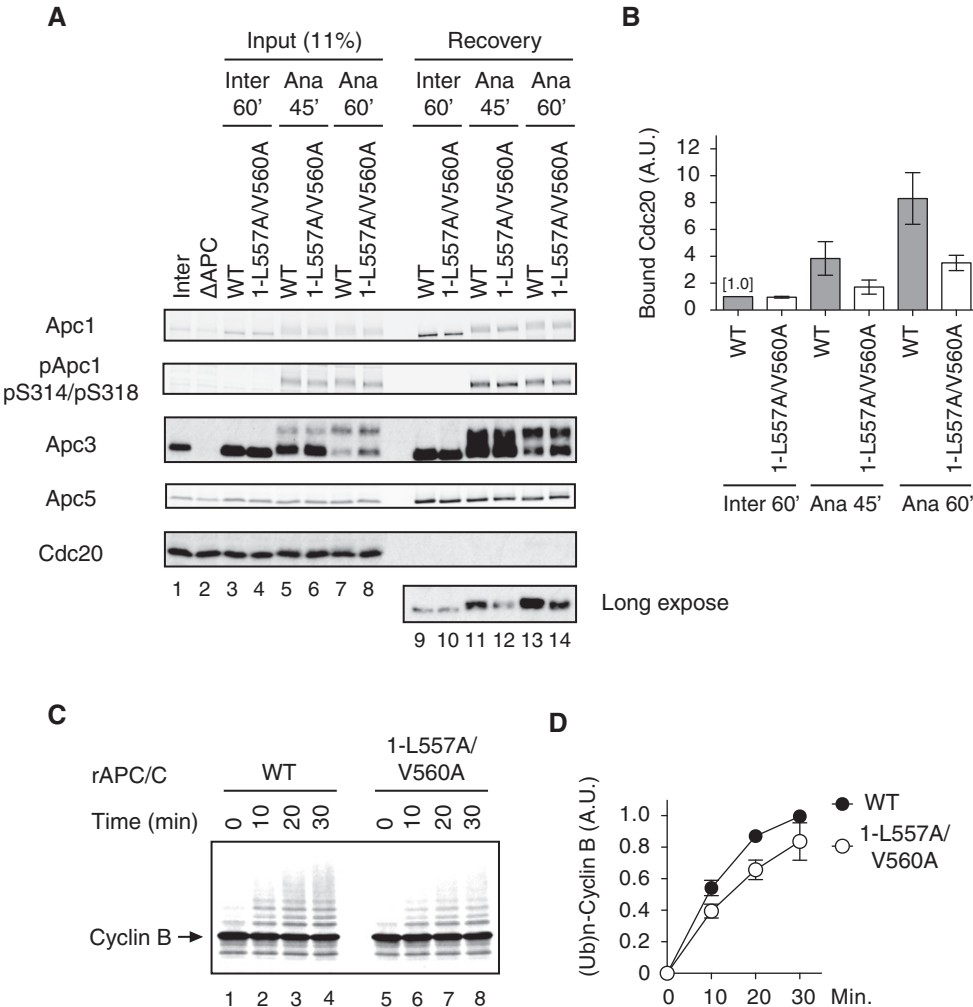

**Figure 2. B56 binding to Apc1-loop$^{500}$ is important for the formation of the active APC/C-Cdc20 complex.**

A  Cdc20 binding assay in *Xenopus* egg extracts. The purified recombinant wild-type (WT) or B56 binding site mutant APC/C (1-L557A/V560A) was incubated with APC/C-depleted (ΔAPC) interphase extract (Inter) or ΔAPC anaphase extract supplemented with CycBΔ167 (Ana) at 23°C for indicated times. The APC/C was recovered with Apc3 monoclonal antibody (AF3.1) beads, and the bound proteins were analysed by SDS–PAGE and immunoblotting with indicated antibodies. pApc1 (pS314/pS318) is a phospho-site-specific antibody that binds only when both S314 and S318 are phosphorylated.

B  Quantification of (A). The intensities of WT control in interphase were arbitrarily set to 1.0. Error bars, SEM from three independent experiments.

C  B56 binding site mutant APC/C (1-L557A/V560A) is less active in ubiquitylation assay than WT APC/C. The purified recombinant WT APC/C or B56 binding motif mutant APC/C (1-L557A/V560A) was incubated with ΔAPC anaphase extract. The recovered APC/C-Cdc20 complex was subjected to ubiquitylation assay using $^{35}$S-labelled cyclin B as a substrate. Samples were taken at indicated time points and analysed by SDS–PAGE and autoradiography. Cdc20 bound to the APC/C is presented in Fig EV2B.

D  Quantification of (C). Error bars, SEM from three independent experiments.

pipeline by which *Xenopus* apo-APC/C is reconstituted by simultaneous co-expression of all 14 subunits in insect cells using the MultiBac system and its activity is examined in APC/C-depleted *Xenopus* egg extracts [12]. We made recombinant *Xenopus* APC/C apo-complexes carrying mutations in Apc1 (Apc1-loop$^{500-L557A/V560A}$ or Apc1-loop$^{500-Δ11}$), and then tested the ability of mutant apo-APC/Cs to bind to the co-activator Cdc20 (Fig 2A and B). When incubated with anaphase extract, APC/C$^{1-L557A/V560A}$ showed a lower affinity for Cdc20 than WT APC/C, although associated Cdc20 levels gradually increased in both cases during incubation (Fig 2A, lanes 11–14). Cdk-dependent phosphorylation in Apc1-loop$^{300}$ (pApc1: pS314/pS318) was observed at almost equal levels in both apo-APC/Cs. In contrast, in

interphase, Cdc20 was able to bind WT APC/C and APC/C$^{1-L557A/V560A}$ at a similar level (Fig 2A, lanes 9 and 10). Consistent results were obtained by using a mutant apo-APC/C harbouring Apc1-loop$^{500-Δ11}$, APC/C$^{1-Δ11}$ (Fig EV2A). These results suggest that Apc1-loop$^{500}$ plays a role in formation of APC/C-Cdc20 complex in anaphase.

We also investigated the activity of WT APC/C and mutant APC/C$^{1-L557A/V560A}$ by *in vitro* ubiquitylation assay after incubation with anaphase extract. The activity of APC/C$^{1-L557A/V560A}$ was lower than that of WT APC/C (Fig 2C and D). This result was consistent with the levels of Cdc20 bound to each apo-APC/C (Fig EV2B). Similarly, APC/C$^{1-Δ11}$ showed lower activity than WT APC/C (Fig EV2C and D). We further examined the activity of mutant

apo-APC/Cs using cyclin destruction assays reconstituted in *Xenopus* egg extracts in which the endogenous APC/C had been depleted before supplementing with the reconstituted APC/C to be tested [12]. Cdc20-dependent cyclin destruction with APC/C[1-L557A/V560A] was slower than that with WT APC/C in anaphase extract (Fig EV3A, upper panel). As a control, we examined cyclin destruction in interphase extract supplemented with Cdh1 and either WT APC/C or APC/C[1-L557A/V560A]. We found that there was no difference in the rate of cyclin destruction in interphase, suggesting that both reconstituted APC/Cs were equally functional (Fig EV3A, lower panel). Similar results were obtained by using the other Apc1-loop[500] B56 binding site mutant, APC/C[1-Δ11] (Fig EV3B). Taken together, we can conclude that the Apc1-loop[500] plays a role in regulating APC/C activity through Cdc20 loading in anaphase. We have previously shown that PP2A is involved in dephosphorylating the N-terminal inhibitory phosphorylation sites of Cdc20, promoting mitotic Cdc20-APC/C complex formation [15]. Thus, it is tempting to speculate that PP2A-B56 bound to the APC/C might contribute towards Cdc20 dephosphorylation. Yet, it is noteworthy that phosphorylation of Apc3 in APC/C[1-L557A/V560A] was slightly delayed although phosphorylation of Cdk1 sites within Apc1-loop[300], which controls engagement of Cdc20 [12–14], occurred to a similar level (Fig 2A). It is possible that the B56 binding site in Apc1-loop[500] may regulate APC/C activation, directly and/or indirectly, through an as-yet-uncharacterised phosphorylation and protein interaction.

## PP2A-B56 bound to Apc1-loop[500] can regulate APC/C activation independently of Apc1-loop[300]

It has been shown that the Apc1-loop[300] acts as a phosphorylation-dependent auto-inhibitory domain, which blocks the engagement of Cdc20, in particular at the C-box and Apc8 interaction for APC/C activation. In order to study the direct role of B56 interaction with Apc1-loop[500] towards the phosphorylation status of Cdc20 rather than Apc3 and Apc1, we first removed this steric auto-inhibitory segment and made apo-APC/Cs harbouring deletion of Apc1-loop[300] (1-ΔL300, deletion of 298–397 in *Xenopus*). This deletion renders APC/C phosphorylation unnecessary for Cdc20-dependent activation. Cyclin destruction assays confirmed that unlike WT APC/C, the APC/C[1-ΔL300] was able to be activated in interphase by Cdc20 (Fig EV4). We combined Apc1-ΔL300 and Apc1-loop[500] mutations and made double mutant APC/Cs in order to assess the impact of APC/C[1-Δ11] or APC/C[1-L557A/V560A] on the binding of Cdc20 independently of Apc1-loop[300] inhibition (Fig 3). As expected, the removal of auto-inhibitory segment, Apc1-loop[300], greatly enhanced the Cdc20 binding to APC/C even in interphase extract (Fig 3A, lanes 7 and 8). Additional mutations of Apc1-loop[500] (APC/C[1-Δ11] or APC/C[1-L557A/V560A]) had no effect on Cdc20 binding in interphase extract (Fig 3A, lanes 9 and 10). In contrast, in anaphase extract, these additional mutations reduced Cdc20 binding (Fig 3B, lanes 8–10). These results demonstrate that the Apc1-loop[500] can promote formation of Cdc20-APC/C complex independently of phosphorylation of the Apc1-loop[300].

## PP2A-B56 mediates dephosphorylation of Cdc20

How does PP2A-B56 bound to Apc1-loop[500] promote the formation of APC/C-Cdc20 complex? It has been shown that PP2A can

antagonise Cdk-dependent inhibitory phosphorylation of Cdc20 at the N-terminal domain (N-Cdc20) and promotes the engagement of Cdc20 with the APC/C for activation [15,29]. In addition, several lines of evidence suggest that PP2A complexes have an inherent preference for phospho-threonine over phospho-serine [29–32]. It has also been reported that Cdk1-dependent inhibitory phosphorylation sites within N-Cdc20 are exclusively threonine (T64, T68 and T79 in *Xenopus* Cdc20) [15,29], whereas Cdk1-dependent stimulatory phosphorylation sites in Apc1-loop[300] are exclusively serine (S314, S318, S335, S344, S358, S380 and S389 in *Xenopus* Apc1) [12]. We showed that dephosphorylation of T79 was rapid and most phosphate was removed within 10 min in anaphase extracts [15]. We investigated the kinetics of phosphate removal from [32]P-labelled Apc1-loop[300] fragment incubated in anaphase extracts (Fig 4A and B). The loss of radioactivity from Apc1-loop[300] was very slow, in great contrast to phosphate removal from T79 on N-Cdc20. Intriguingly, Apc1-loop[300-7T] fragment in which all conserved serine Cdk sites had been replaced by threonine was rapidly dephosphorylated with similar kinetics to N-Cdc20 (Fig 4A and B). We therefore hypothesised that PP2A-B56 bound to Apc1-loop[500] might be involved in dephosphorylation of Cdc20, but not Apc1-loop[300], and promotes the formation of Cdc20-APC/C complex. If this hypothesis is correct, a non-phosphorylatable Cdc20 mutant (Cdc20-5A; S50A, T64A, T68A, T79A and S114A), which is free from Cdk-dependent inhibitory phosphorylation [15], should become insensitive to B56 binding site mutations and load to the APC/C. We examined whether Cdc20-5A efficiently binds to the APC/C regardless of the presence of B56 binding site mutations in the Apc1-loop[500] (Fig 4C and D). As expected, in the case of Cdc20-WT, the ability to bind the APC/C harbouring B56 binding site mutations (APC/C[1-L557A/V560A]) was decreased, compared with WT APC/C (Fig 4C, lanes 7 and 8). However, for Cdc20-5A, the effect of the B56 binding mutation (Apc1-L557A/V560A) was significantly alleviated and Cdc20-5A efficiently bound APC/C[1-L557A/V560A] (lanes 9 and 10). In support of this, Cdc20-5A was able to bind equally to both WT APC/C and APC/C[1-Δ11], another B56 binding site mutant APC/C (Fig EV5). It should be noted that the levels of Cdk1 phosphorylation of Apc1-loop[300] (pApc1: pS314/pS318) are similar in both WT and B56 binding site mutant APC/C in anaphase extract (Figs 2A and EV2A, lanes 11–14), implying that PP2A-B56 recruited by Apc1-loop[500] does not dephosphorylate Cdk1 sites of Apc1-loop[300]. These results suggest that PP2A-B56 bound to Apc1-loop[500] can dephosphorylate N-Cdc20 and promote Cdc20-APC/C complex formation.

Finally, we wanted to examine the specificity of PP2A-B56 towards key phosphorylation sites on Cdc20 and the APC/C. Thus, we have tested whether purified PP2A-B56γ is capable of dephosphorylating the inhibitory threonine residues of N-Cdc20 more efficiently than Apc1-loop[300]. Active recombinant PP2A-B56 was expressed and purified from insect cells and mixed with mitotically phosphorylated Cdc20 or APC/C as substrates for dephosphorylation, which was monitored by immunoblotting with anti-pT79 (Cdc20) or anti-pS314/S318 (Apc1-loop[300]). Phosphorylation of Cdc20 T79 was significantly reduced upon addition of PP2A-B56γ, whereas phosphorylation of Apc1-loop[300] remained stable (Fig 4E, lanes 5–8). At a higher concentration of PP2A-B56γ, Apc1-loop[300] was dephosphorylated to some extent, but Cdc20 pT79 was again more efficiently dephosphorylated (Fig 4E, lanes 9–12). This supports our model (Fig 4F) that PP2A-B56 bound to the APC/C

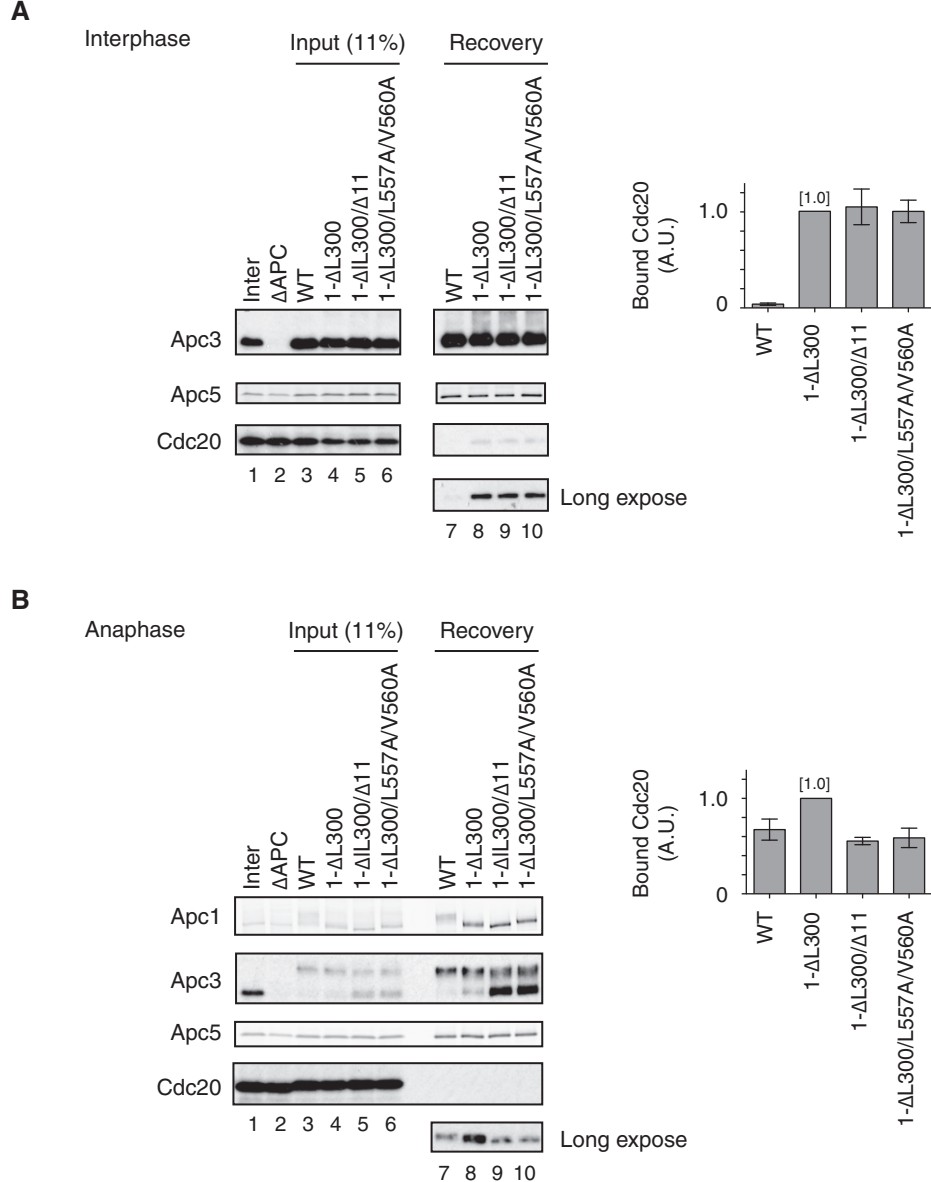

**Figure 3. Apc1-loop500 regulates APC/C activation independently of phosphorylation of Apc1-loop300.**

A  (left panel) Cdc20 binding to APC/C loop domain mutants in interphase extract. The purified recombinant WT or Apc1 mutant APC/Cs (1-ΔL300, 1-ΔL300/Δ11 or 1-ΔL300/L557A/V560A) were incubated with ΔAPC interphase extract at 23°C for 1 h. The APC/C was recovered with Apc3 monoclonal antibody (AF3.1) beads, and the bound proteins were analysed by SDS–PAGE and immunoblotting with indicated antibodies. (right panel) Quantification of bound Cdc20. The intensities of 1-ΔL300 control were arbitrarily set to 1.0. Error bars, SEM from three independent experiments.

B  (left panel) Cdc20 binding to APC/C loop domain mutants in anaphase extracts. Same as (A) but ΔAPC anaphase extract was used and incubated at 23°C for 50 min in anaphase extract. (right panel) Quantification of bound Cdc20. The intensities of 1-ΔL300 control were arbitrarily set to 1.0. Error bars, SEM from three independent experiments.

enables efficient dephosphorylation of Cdc20, but not the APC/C, resulting in Cdc20 association with the APC/C in mitosis and initiating APC/C-dependent ubiquitylation.

Thus, our study reveals for the first time the presence of a binding site for PP2A-B56 within the APC/C subunit and we propose a PP2A-B56-mediated mechanism controlling Cdc20-APC/C formation in mitosis. For the maintenance of genome stability, accurate sister chromatid separation, alongside faithful DNA replication, is arguably one of the most important events during the cell cycle. This

separation is triggered by tightly regulated activation of the APC/C and subsequent ubiquitin-mediated proteolysis of the anaphase inhibitor securin, which liberates the protease separase, which in turn cleaves the kleisin subunit of the cohesin complex that physically holds sister chromatids together. Cdk1-dependent phospho-regulation is clearly central for APC/C regulation but is not fully understood. For example, although the APC/C and Cdc20 form an active complex in mitosis, phospho-regulation of APC/C and Cdc20 occurs in an opposing manner; the APC/C needs to be phosphorylated,

whereas Cdc20 needs to be dephosphorylated for its engagement and subsequent C-box-dependent activation of the APC/C [15]. PP2A has been shown to be important for dephosphorylation of APC/C and Cdc20 [15,28,29]. It has also been reported that in mitotic exit after cyclin B degradation, PP2A-B55 is crucial and preferentially dephosphorylates phospho-threonine over phospho-serine residues [29–32]. However, until now, there has been no direct evidence to show how Cdc20 is dephosphorylated at mid-mitosis when the activity of PP2A-B55 is strictly downregulated via the Greatwall-ENSA pathway. Cdk phosphorylation clearly stimulates

the affinity of Apc1-loop[500] towards PP2A-B56 (Fig 1E and F). Furthermore, our *in vitro* experiments using purified PP2A-B56, Cdc20 and APC/C demonstrate that PP2A-B56 can efficiently dephosphorylate the N-terminal inhibitory domain of Cdc20 (Fig 4E). Consistently, reconstituted mutant apo-APC/C complexes deficient in B56-APC/C interactions show less activity than WT APC/C (Figs 2 and EV2). These results demonstrate that PP2A-B56, instead of PP2A-B55, might be responsible for the initial dephosphorylation of Cdc20 promoting the formation of an active APC/C-Cdc20 complex required for anaphase onset.

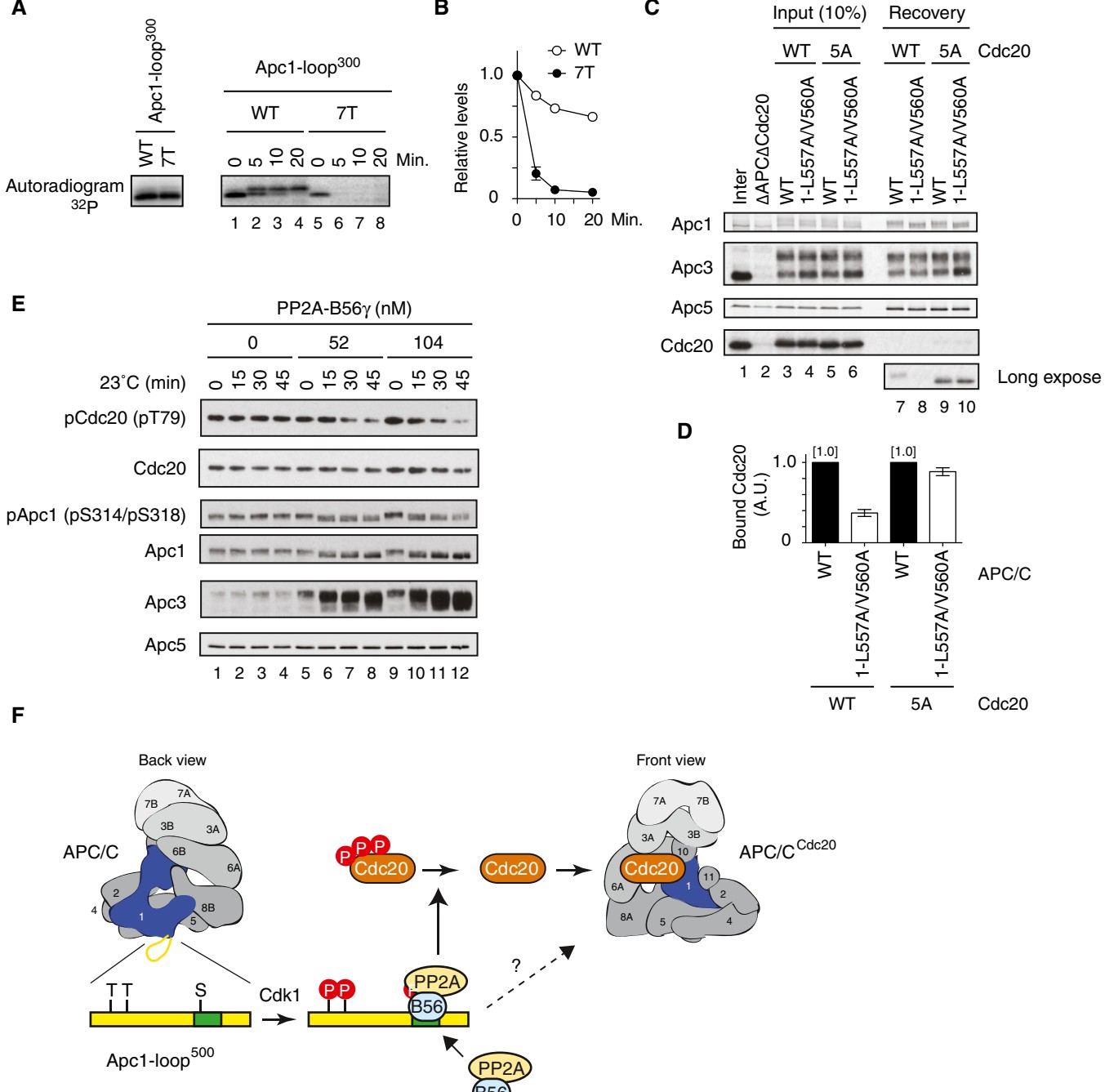

**Figure 4.**

**Figure 4.  PP2A-B56 bound to Apc1-loop$^{500}$ promotes APC/C-Cdc20 complex formation through dephosphorylation of N-terminal Cdc20.**

A  (left panel) Cdk-dependent *in vitro* kinase assay of Apc1-loop$^{300}$. 3xFlag-tagged WT or 7T Apc1-loop$^{300}$ fragment was incubated with Cdk2-cyclin A in the presence of [γ-$^{32}$P]-ATP at 30°C for 30 min, and separated by SDS–PAGE and detected by autoradiography. (right panel) $^{32}$P-phosphorylated WT or 7T Apc1-loop$^{300}$ fragment was incubated in anaphase extract, and removal of radioactivity was analysed by SDS–PAGE followed by autoradiography.

B  Quantification of removal of radioactivity in (A). Error bars, SEM from three independent experiments.

C  Cdc20-5A efficiently binds to the B56 binding site mutant APC/C. The purified recombinant WT APC/C or Apc1-loop$^{500}$ mutant APC/C (1-L557A/V560A) was incubated with WT Cdc20 or non-phosphorylatable Cdc20 mutant (5A) in ΔAPCΔCdc20 anaphase extract at 23°C for 55 min. The APC/C was recovered with Apc3 monoclonal antibody (AF3.1) beads, and the bound proteins were analysed by SDS–PAGE and immunoblotting with indicated antibodies.

D  Quantification of (C). The intensities of WT APC/C control were arbitrarily set to 1.0. Error bars, SEM from three independent experiments.

E  The APC/C and Cdc20 immunoprecipitated from CSF extract were incubated in the presence of a range of concentrations or absence of PP2A-B56γ at 23°C. Samples were taken at the indicated time points and analysed by SDS–PAGE and immunoblotting with antibodies, including phospho-site-specific antibodies for pCdc20 (pT79) and pApc1 (pS314/pS318).

F  A model for Apc1-loop$^{500}$-mediated Cdc20-APC/C complex formation in mitosis. Apc1-loop$^{500}$ is phosphorylated by Cdk1 and binds to PP2A-B56 in anaphase. PP2A-B56 dephosphorylates inhibitory phosphorylation sites in N-Cdc20 and promotes the formation of active APC/C-Cdc20 complex. Apc1-loop$^{500}$ may control phosphorylation of other APC/C subunits and possibly interacting proteins although the mechanisms involved remain elusive. The indicated numbers on the schematic view of the APC/C (the back view is rotated by 180° around the vertical axis from the front view) represent APC/C subunits.

The B56 binding site is located within a disordered loop domain (Apc1-loop$^{500}$) in the N-terminal wheel-like WD40 domain of Apc1 (Appendix Fig S1A). It should be noted that the same WD40 domain includes another disordered loop domain Apc1-loop$^{300}$, which acts as a phosphorylation-dependent auto-inhibitory segment, on the opposite side of the WD40 domain (Appendix Fig S1B). Our experiments using purified PP2A-B56 show a preference for phospho-threonine in N-Cdc20 over phospho-serine residues in the Apc1-loop$^{300}$, which is geographically closer to Apc1-loop$^{500}$ than Cdc20 associated with the APC/C. Strikingly, in anaphase extracts phospho-serine residues in the Apc1-loop$^{300}$ are barely dephosphorylated unlike the threonine substitutions or phospho-threonine in N-Cdc20 (Fig 4A and B) [15]. These findings suggest that the phospho-threonine preference of PP2A-B56 is a key feature in the control of mitotic progression through APC/C-Cdc20 complex formation and APC/C-dependent degradation. It is worth noting that when mitotic exit phosphatases such as PP2A-B55 and PP1 (Cdc14 in budding yeast) become active after cyclin destruction, the APC/C is fully dephosphorylated and is no longer responsive to Cdc20. At this point, a structurally related co-activator, Cdh1, takes over and prompts the APC/C to degrade APC/C substrates until the onset of S phase. Interestingly, it remains to be elucidated exactly how Cdh1 can activate the unphosphorylated APC/C by displacing Apc1-loop$^{300}$ [12–14].

How is timely recruitment of PP2A-B56 regulated in mitosis? PP2A-B56 binding to Apc1-loop$^{500}$ is dependent upon phosphorylation of the Cdk1 sites in Apc1-loop$^{500}$ (Fig 1E and F), suggesting that dephosphorylation of Cdc20 may be accelerated when APC/C is fully phosphorylated in mid-mitosis. This correlates with the recent finding that phosphorylation in the vicinity of the B56 binding motif can increase PP2A-B56 binding [25–27]. Notably, the Apc1-loop$^{500}$ sequence and Cdk sites are highly conserved through evolution (Fig 1A), so the disordered loop domain may be monitoring stages in the cell cycle by sensing Cdk1 activity. PP2A-B56 recruited onto phosphorylated Apc1-loop$^{500}$ might increase the local concentration of dephosphorylated Cdc20 around phosphorylated APC/C, promoting the formation of APC/C-Cdc20 complex (Fig 4F). It has been recently reported that at kinetochores, depending on microtubule attachment, APC/C activation might be regulated via Cdc20 dephosphorylation by kinetochore-localised PP1 [33]. PP2A-B56 activity is also needed to silence the SAC [23,24]. In addition, PP1 and PP2A-B56 have recently been shown to exploit inverse phospho-dependencies at the kinetochore because phosphorylation of PP1- or PP2A-interacting motif has an opposite impact on the phosphatases,

inhibitory to PP1 vs. enhancement to PP2A recruitment [34]. It is possible that B56 binding to Apc1-loop$^{500}$ may be more important in a situation or a specific area where a mitotic kinase is high and PP1 is less responsive. In more general terms, PP2A-B56 recruited on APC/C may also facilitate dephosphorylation of Cdc20 in a specific location, which may trigger a positive feedback whereby dephosphorylation of Cdc20 and APC/C-Cdc20 complex formation occur in a spatially and temporally regulated manner. Investigation of living cells may shed light on such roles in the regulation and subcellular localisation of the APC/C itself and neighbouring proteins. Understanding the role of PP2A-B56 recruited on Apc1-loop$^{500}$ on SAC silencing/MCC disassembly is similarly intriguing. Our study demonstrates that Apc1-loop$^{500}$ promotes APC/C-Cdc20 complex formation under normal conditions and is not dependent upon the SAC as it is not active in *Xenopus* egg extracts. Yet, whether it regulates SAC silencing/MCC disassembly remains to be investigated.

In summary, we find that a disordered loop domain (Apc1-loop$^{500}$) plays an important role in dephosphorylating Cdc20 by directly recruiting PP2A-B56, thereby allowing productive Cdc20-APC/C complex formation in mitosis. The Apc3-loop mediates efficient phosphorylation of Apc3 and Apc1 by directly recruiting Cks-Cdk1-cyclin B, whereas the Apc1-loop$^{300}$ acts as a phosphorylation-mediated conformational switch, permitting association with Cdc20. It seems that flexible and disordered loop domains within the APC/C play a central role in the dynamic regulation of the APC/C, in particular through post-translation modifications (PTMs). This appears to be the case not only for the APC/C but also for other multi-protein complexes and cellular regulation pathways, e.g. RNA metabolism. Intriguingly, a number of recently identified RNA binding proteins bind to RNAs via disordered loop regions [35]. More unexpected modes of binding or critical regulation of key proteins in our body might be uncovered by studying unstructured disordered loop domains.

# Materials and Methods

### Plasmids and antibodies

Antibodies used are as follows: anti-Apc1 (RbAb 4853, 1:100), Apc3/Cdc27 (1:200; BD Transduction Laboratories), Apc5 (RbAb 3445, 1:500), Cdc20 (MAb BA8, 1:100), maltose binding protein (MAb R29, 1:500 except for Fig 1E, 1:5,000), Apc1 phosphopeptide

(pS314-pS318) antibody (RbAb 25312, 1:50) [12], Cdc20 phospho-peptide (pT79) antibody (MAb BT2.1, 1:25) [15], and Flag-M2-HRP (for Fig 1E, A8592, Sigma, 1:1,000). AF3.1 and B60 were used for immunoprecipitation or immunodepletion of APC/C and Cdc20, respectively [36]. Plasmids for the expression of *X. laevis* PP2A regulatory B subunits and PP2A (Aα, B56γ and Cβ) subunits are gifts from Drs N. Sagata [37] and S. Mochida [17], respectively.

## Preparation of *Xenopus* egg cell-free extracts

Meiotic metaphase II-arrested (CSF) *X. laevis* egg extracts were prepared as described [38]. To prepare interphase extracts, CSF extracts were incubated at 23°C for 1.5 h in the presence of 0.4 mM CaCl$_2$ and 10 μg/ml cycloheximide, a protein synthesis inhibitor. Anaphase extracts were prepared by adding non-degradable GST-fused *Xenopus* cyclin BΔ167 (a truncated form of cyclin B lacking the N-terminal 167 amino acids) to interphase extracts and incubating for 30–60 min at 23°C [12]. APC/C-depleted (ΔAPC) or Cdc20-depleted (ΔCdc20) extracts were prepared as reported previously [36,39].

## Expression and purification of recombinant APC/C

Expression and purification of recombinant APC/Cs were performed as described previously [12,40]. Briefly, two baculoviruses carrying intact *Xenopus* APC/C genes and TEV-cleavable tandem Strep II-tag fused to Apc6 at C-terminus (Apc6-strep) were generated by MultiBac system (baculovirus 1: Apc1, Apc2, Apc10 and Apc11; baculovirus 2: Apc3, Apc4, Apc5, Apc6-strept, Apc7, Apc8, Apc12, Apc13, Apc15 and Apc16). Mutant APC/Cs were generated by PCR-based mutagenesis, and mutation sites were confirmed by sequencing. To express APC/C complex, High Five insect cells (Invitrogen) at a cell density of $1.5 \times 10^6$ were co-infected with the two recombinant baculoviruses at an MOI (multiplicity of infection) of 1 for each virus and incubated at 27°C for 48 h with shaking (150 rev/min). The cells were harvested, frozen in liquid nitrogen and stored at −80°C. The recombinant APC/Cs were purified with Strep-Tactin beads and further affinity-purified by Dynabeads Protein A conjugated to anti-Apc3 monoclonal antibody (MAb AF3.1). The APC/Cs bound to beads were flash-frozen and stored at −80°C.

## Construction and purification of Apc1-Loop$^{500}$, Apc1-Loop$^{300}$ and B56γ

The Apc1-Loop$^{500}$, spanning amino acid 515–584 of *Xenopus tropicalis* Apc1, was fused with PreScission protease-cleavable maltose binding protein (MBP) at the N-terminus and a TEV-cleavable 6xHis at the C-terminus, and subcloned into pET vector. Mutants were generated by PCR-based mutagenesis and subcloned into pET vector as for WT, and mutation sites were confirmed by sequencing. The Apc1-Loop$^{300}$, spanning amino acids 294–399 of *X. tropicalis* Apc1, was fused with a PreScission protease-cleavable 3xFlag-tag at the N-terminus and a TEV-cleavable 6xHis at the C-terminus, and subsequently subcloned into pET vector. For its 7T variant (Apc1-loop$^{300-7T}$), cDNA encoding for the same region but containing threonine substitutions of all the conserved serine Cdk sites (S314T, S318T, S335T, S344T, S358T, S380T and S389T) was synthesised (Integrated DNA Technologies, BVBA) and subcloned into the pET vector as for WT. In both WT and mutant vectors, the

Cdk sites (SP or TP) were confirmed by sequencing prior to bacterial expression. *Xenopus laevis* B56γ was fused with PreScission protease-cleavable 3xFlag at the N-terminus and a TEV-cleavable 6xHis at the C-terminus, and subcloned into pET vector. After sequencing confirmation, the resultant plasmids were introduced into BL21-CodonPlus (DE3) and the fusion proteins were expressed at 37°C for 1 h in the presence of 1 mM IPTG. The cells were lysed by 0.3 mg/ml lysozyme and sonicated in lysis buffer (20 mM HEPES-NaOH pH 7.9, 500 mM NaCl, 5 mM EGTA, 10 μg/ml leupeptin, 10 μg/ml pepstatin A, 10 μg/ml chymostatin, 0.1% Triton X-100 and 10 mM imidazole). The proteins were purified from clarified lysate using Ni-NTA agarose beads (Qiagen).

## Kinase assays

MBP-fused WT or 3A Apc1-loop$^{500}$ fragment (6 μg) was incubated with Cdk2-cyclin A in the presence of [γ-$^{32}$P]-ATP at 23°C for 10 min, and separated by SDS–PAGE and detected by autoradiography (for Fig 1D). 3xFlag-tagged WT or 7T Apc1-loop$^{300}$ fragment (140 ng) was incubated with Cdk2-cyclin A in the presence of [γ-$^{32}$P]-ATP at 30°C for 30 min, and separated by SDS–PAGE and detected by autoradiography (for Fig 4A).

## PP2A regulatory B subunits and Apc1-loop$^{500}$ binding assays

For binding assay with [$^{35}$S]-labelled PP2A B subunits and Apc1-Loop$^{500}$ fragments, N-terminally 6xHis-2xFlag-tagged *X. laevis* PP2A regulatory B subunits (B56γ, B55δ or B″/PR70) [37] were labelled with [$^{35}$S]methionine (Hartmann Analytic, UK) in a coupled *in vitro* transcription–translation system (Promega, UK). Purified Apc1-Loop$^{500}$ fragment proteins were bound to amylose beads (New England Biolabs) by incubating at 4°C for 0.5–1 h. Beads were washed with Tris-NaCl buffer [20 mM Tris–HCl pH 8.0 and 200 mM NaCl] and XB$^{CSF}$ buffer [10 mM HEPES-KOH pH 7.8, 50 mM sucrose, 100 mM KCl, 2 mM MgCl$_2$ and 5 mM EGTA] and incubated with $^{35}$S-labelled PP2A B subunits in interphase extract in the presence or absence of cyclin BΔ167 at 23°C for 60 min, separated from extract on Micro Bio-Spin columns (Bio-Rad), and washed once with XB$^{CSF}$ buffer and then twice with XB$^{CSF}$ buffer containing 0.01% NP-40. The bound proteins were eluted with SDS sample buffer and analysed by SDS–PAGE and autoradiography. For Cdk phosphorylation-dependent B56 loading assays, MBP-fused WT or 3A Apc1-loop$^{500}$ fragment (12 μg) was first bound to amylose resin and then incubated at 30°C for 60 min in 20 μl of XB$^{CSF}$ buffer containing 1 mM ATP and 4 mM MgCl$_2$ in the presence or absence of Cdk2-cyclin A. Beads with MBP peptides (+/− kinase) were washed with XB$^{CSF}$ buffer containing 0.01% NP-40 and incubated in 20 μl of the same buffer containing purified 3xFlag-tagged B56γ (4 μg) at 23°C for 30 min. The beads were separated on Micro Bio-Spin columns (Bio-Rad), washed once with XB$^{CSF}$ buffer and then twice with XB$^{CSF}$ buffer containing 0.01% NP-40. The bound proteins were eluted with SDS sample buffer and analysed by SDS–PAGE and immunoblotting.

## Immunoprecipitation of APC/C

The APC/C was immunoprecipitated using Apc3 MAb (AF3.1) immobilised Dynabeads Protein A beads. The bound proteins were

washed twice with XB$^{CSF\_HS}$ [XB$^{CSF}$ containing 500 mM KCl and 0.01% NP-40], eluted with SDS sample buffer and analysed by SDS–PAGE and immunoblotting.

### Cell-free destruction assay

Destruction assays were performed as described previously [12]. Substrates were labelled with [$^{35}$S]methionine (Hartmann Analytic, UK) in a coupled *in vitro* transcription–translation system (Promega, UK), and destruction assays were carried out using *Xenopus* egg cell-free extracts (anaphase or interphase extracts). The samples were taken at the indicated time points and analysed by SDS–PAGE and autoradiography. The images were analysed using ImageJ (NIH, USA).

### Ubiquitylation assay

Ubiquitylation assays were performed as described [12]. Recombinant APC/Cs were incubated with *Xenopus* egg cell-free extracts in the presence of non-degradable GST-fused *Xenopus* cyclin BΔ167 at 23°C for 60 min and immunoprecipitated using Apc3 MAb (AF3.1)-immobilised Dynabeads Protein A. The beads were washed twice with XB$^{CSF\_HS}$ and once with Ub buffer (20 mM Tris–HCl pH 7.5, 100 mM KCl, 2.5 mM MgCl$_2$ and 0.3 mM DTT). The resultant APC/C-Cdc20 complexes were incubated at 23°C in 20 μl of buffer (20 mM Tris–HCl pH 7.5, 100 mM KCl, 2.5 mM MgCl$_2$, 2 mM ATP and 0.3 mM DTT) containing 0.05 mg/ml E1, 0.025 mg/ml UbcX, 0.75 mg/ml ubiquitin, 1 μM ubiquitin aldehyde, 150 μM MG132 and 2 μl of $^{35}$S-labelled cyclin B (fission yeast Cdc13). The reactions were stopped at the indicated time points with SDS sample buffer and analysed by SDS–PAGE and autoradiography.

### Cloning and expression of PP2A-B56γ

Three PP2A subunit genes (Aα, B56γ and Cβ) were PCR-amplified and cloned into pOENmyc vector with polH promoter and SV40 terminator for Cβ or p10 promoter and HSVtk terminator for Aα and B56γ. PreScission protease-cleavable GST-tag or His-tag was fused to Aα or B56γ at the N-terminus, respectively. These genes were further cloned into MultiBac vectors creating one pF1 vector-derivative pFUBB carrying Cβ at MUM1 site and His-B56γ at MUM2 site. A pU1 vector-derivative pUUBB carrying Aα at MUM2 site was also created. After sequence confirmation, generation of baculovirus and the expression of PP2A-B56γ complex were performed as described previously [12] except that the baculovirus was used at MOI of 2 for expression.

### Purification of PP2A-B56γ

Cell pellets were thawed on ice and resuspended in PP2A lysis buffer [50 mM Tris–HCl pH 7.5, 150 mM NaCl, 0.5 mM DTT, 0.1% Tween-20, 1 mM EDTA, 1 mM EGTA, 5% glycerol, 10 μg/ml leupeptin, 10 μg/ml pepstatin A, 10 μg/ml chymostatin and 30 units/ml benzonase (Novagen)] and lysed by sonication. The lysate was centrifuged at 18,800 $g$ for 20 min, and the supernatant was centrifuged again for 20 min. The cleared lysate was incubated with GSH beads (GE Healthcare) at 4°C for 1 h. The beads were washed twice with PP2A wash buffer 1 (50 mM Tris–HCl pH 7.5, 150 mM NaCl, 0.5 mM DTT, 0.1% Tween-20, 1 mM EDTA and 1 mM EGTA) and once with PP2A wash buffer 2 (50 mM Tris–HCl pH 7.5, 150 mM NaCl, 0.5 mM DTT, 1 mM EDTA and 5% glycerol) and suspended into PP2A wash buffer 2 containing PreScission protease and incubated at 4°C for 3.5 h with gently mixing. After a brief centrifugation, the supernatant was incubated with Ni-NTA at 4°C for 30 min with gentle mixing, and the beads were washed three times with His washing buffer (20 mM Tris–HCl pH 7.5, 150 mM NaCl, 3 mM DTT, 0.01% Tween-20, 5% glycerol and 20 mM imidazole), and the bound proteins were eluted by His elution buffer (20 mM Tris–HCl pH 7.5, 150 mM NaCl, 3 mM DTT, 0.01% Tween-20, 5% glycerol and 200 mM imidazole). All purification steps were performed at 4°C.

### Phosphatase assay

APC/C and Cdc20 were immunoprecipitated from CSF extract using anti-APC/C (AF3.1) and anti-Cdc20 (B60) antibody-immobilised Dynabeads Protein A. The beads were washed twice with XB$^{CSF\_HS}$ without EGTA and once with XB$^{CSF}$ buffer containing 0.01% NP-40 without EGTA, resuspended into XB$^{CSF}$ buffer containing 0.01% NP-40 without EGTA and incubated with purified PP2A-B56γ at 23°C. The reactions were stopped at the indicated time points with SDS sample buffer and analysed by SDS–PAGE and immunoblotting. For time course phosphate removal experiment (Fig 4A), 3xFlag-tagged WT or 7T Apc1-loop$^{300}$ [γ-$^{32}$P]-ATP phosphorylated by Cdk2-cyclin A was incubated in anaphase extract at 23°C. The samples were taken at the indicated time points and analysed by SDS–PAGE and autoradiography.

### Statistical analyses

Statistical analyses were performed in GraphPad Prism v6.0. Quantification data are presented as the mean ± SEM from three independent experiments.

**Expanded View** for this article is available online.

### Acknowledgements

We thank N. Sagata for plasmids expressing PP2A regulatory B subunits; S. Mochida for plasmids expressing PP2A subunits; H. Labit for *X. laevis* PP2A-B56γ baculovirus; S. Darling for Apc1-loop$^{300}$ fragments; J. Nilsson for information about a B56 binding SLiM; the staff at the UCL Biological Services Unit for taking care of the *Xenopus* colony at UCL; and M. Grimaldi and members of the Yamano laboratory for helpful discussions and critical reading of the manuscript. This work was supported by the Medical Research Council (MR/M010899/1), BBSRC (BB/N008383/1) and the Wellcome Trust (205150/Z/16/Z).

### Author contributions

KF and HY designed research. KF and HY performed research and analysed data. KF and HY wrote the paper. HY acquired funding and supervised research.

### Conflict of interest

The authors declare that they have no conflict of interest.

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
