## [Review Process File · EMBO Reports]

PP2A-B56 binds to Apc1 and promotes Cdc20 association with the APC/C ubiquitin ligase in mitosis

Kazuyuki Fujimitsu and Hiroyuki Yamano

Review timeline:

Submission date:	16 May 2019
Editorial Decision:	19 June 2019
Revision received:	26 September 2019
Editorial Decision:	28 October 2019
Revision received:	31 October 2019
Accepted:	11 November 2019

Editor: Deniz Senyilmaz-Tiebe

Transaction Report:

1st Editorial Decision

19 June 2019

Thank you for the submission of your research manuscript to our journal. We have now received two referee reports that are copied below.

As you will see, both referees express interest in the findings demonstrating the role of PP2A in initiating the dephosphorylation of Cdc20. However, they also raise some concerns that need to be addressed for publication here.

Given these constructive comments, we would like to invite you to revise your manuscript with the understanding that the referee concerns (as detailed above and in their reports) must be fully addressed and their suggestions taken on board. Please address all referee concerns in a complete point-by-point response. Acceptance of the manuscript will depend on a positive outcome of a second round of review. It is EMBO reports policy to allow a single round of revision only and acceptance or rejection of the manuscript will therefore depend on the completeness of your responses included in the next, final version of the manuscript.

REFeree REPORTS

Referee #1:

In the submitted ms, Fujimitsu and Yamano investigate the regulation of Cdc20 binding to the APC. Previous work revealed that Cdk1 mediated phosphorylation of APC3 recruits Cdk1 via Cks to APC3, which causes the phosphorylation of APC1. Phosphorylated APC1 undergoes a conformational switch liberating the C-box binding site resulting in Cdc20 binding and APC activation. Additional data revealed that Cdc20 is negatively phosphorylated at its C-terminus by Cdk1 and that a strong preference of PP2A-B55 for phosphorylated threonine residues ensures that Cdc20 is dephosphorylated, while APC subunits are still phosphorylated. Based on these findings, Fujimitsu and Yamano investigate the precise role of PP2A in initiating the dephosphorylation of

Cdc20. Using the SLiM for B56 (LSPVPE), the authors identify a putative B56 binding motif in a predicted disordered loop of APC1 (APC1-loop500). Pull-down assays revealed that WT loop500, but not a variant lacking the B56 binding motif or carrying two mutations in the B56 binding motif, or a variant carrying 3A mutations, co-precipitated ectopic B56 from anaphase extract. Next, using their pipeline of APC reconstitution in extract depleted of endogenous APC the authors want to make the point that the APC/C with the two mutations in the B56 binding motif (L557A/V560A) binds less efficiently to Cdc20 than wt APC/C. Notably, APC1 phosphorylation was comparable in both APC/Cs. Ubiquitylation assays were used by the authors to confirm that the activity of APC/C (L557A/V560A) was lower than wt APC/C. Next, the authors employ degradation assays to show that the rate of cyclin B degradation seemed to be higher when WT APC/C was used compared to the B56 binding mutant. To discriminate between B56 binding being important for APC1/APC3 dephosphorylation vs. Cdc20 dephosphorylation, the authors use a variant of APC1 lacking the autoinhibitory segment. As expected, Cdc20 binds more efficiently to the APC/C lacking the autoinhibitory segment in APC/C1. Additional mutation of APC1-loop500 reduced Cdc20 binding in anaphase extract. To confirm that B56 bound to APC1-loop500 dephosphorylates Cdc20, the authors use the Cdc20-5A mutant. This mutant seemed to bind equally well to the APC/C with wt APC1 and APC1 carrying mutations in the B56 binding motif. From these results, the authors conclude that B56 recruited to APC1-loop500 dephosphorylates Cdc20. Using phospho-specific Ab, the authors want to demonstrate that B56 has a preference for Cdc20 over APC1.

The presented data are of great importance for the cell cycle community because it addresses an important point, i.e. how is the activation of Cdc20 mediated under conditions when Cdk1 activity levels are still high. This reviewer therefore thinks that the submitted ms should be published. However, this reviewer is not always convinced by the presented data and the conclusions drawn by the authors. The authors should address the following points:

In general, as outlined below the observed effects are very mild and sometimes even difficult to see. Therefore, to support the conclusions it is important that the authors quantify all experiments and show statistics from several experimentally independent experiments.

1. Fig. 1C: Specific binding over background binding of MBP-APC1-515-584 to WT APC1 is rather weak. The same applies for Fig. 3.
2. The same applies to EV2B and EV2C: This reviewer has a hard time to see a difference in ubiquitylation efficiency of cyclin B and binding of Cdc20 to WT vs. Delta11 APC1.
3. The same applies to Fig. EV3: The effects on the degradation of cyclin B are marginal.
4. Fig4A: It seems that the Cdc20-5A mutant still binds stronger to the wt than the mutant APC/C. In Fig EV5, the Cdc20-5A mutant seems to bind equally to both wt and mutant APC/C. Could the authors comment on this and provide quantifications?
5. There is a strong effect of WT APC and the B56 binding deficient mutant on APC3 phosphorylation levels, e.g. EV2C. How the authors explain this. Please, comment.
6. Figure 4B: the authors know the exact binding site of B56 on the APC/C, so the binding mutant would be a good control in the in vitro phosphatase assay.

Minor Points:

- Page 3: the authors state that "there can be no APC/C activation without Cdk1..." and later that "...Cdh1 can activate both phosphorylated and unphosphorylated APC/C." These statements seem to be contradictory.
- Page 4: the authors use the term "short-conserved". Do they mean "short and conserved"?
- Page 4: the authors write that Cdk1 activity is high in anaphase. Is this specific for the way the Xenopus extracts used here are prepared? In normal mitotic cell divisions, Cdk1 activity is decreasing substantially in (early) anaphase.
- Page 4: the part where the authors talk about the activities of PP2A enzymes in cell cycle. It is written that PP2A-B55 is vital for mitotic exit and later that it is inactive in mitosis, which is somewhat contradictory/imprecise. In addition, the Nilsson lab has shown that B55-PP2A is

important for Cdc20 dephosphorylation at mitotic exit once it is activated. Maybe this should be mentioned here as well.

- Figure 1A: In my printout the lines with the sequences are shifted so that corresponding amino acids are not in a column and the green box is not on the motif.
- Page 6 and Figure 1C: the authors show that the 3A mutant is compromised in B56 recruitment and suggest that this due to the absence of phosphorylation (by Cdk1). Is there more evidence that these sites (especially the serine in the B56-binding motif) are indeed phosphorylated (by Cdk1) in this context?
- Page 10: it is unclear why a Cdc20 with 5 alanine mutations was used when before it was mentioned that only 3 threonine residues are important. I also couldn't find the information about the identity of the mutated amino acids

Referee #2:

Fujimitsu and Yamano investigate the interplay between PP2A-B56 and APC/C-Cdc20 to determine how this may regulate APC/C activity during mitosis. They show that PP2A-B56 binds in a phospho-dependent manner to a canonical LxxIxE binding motif in a disordered loop in APC1. They examine the function of this binding using purified APC/C from a multibac system (+/- B56 binding site) to reconstitute APC/C-depleted *Xenopus* extracts, showing an effect on APC/C-Cdc20 loading and APC/C activity. These effects are independent of the activating Cdk phosphorylations in APC1 and are due to Cdc20 dephosphorylation because a non-phosphorylatable Cdc20 does not require APC1-B56 interaction. Finally, the author suggests that the reason that B56 preferentially dephosphorylates CDC20 and not APC1 is that Cdc20 contains threonine residue that are better substrates for PP2A than the serine residue in APC1- showing evidence for these sites being dephosphorylated with differential kinetics

Overall this is a nice focused paper on a topic that could have important implications. On the downside, I was not overly convinced with some of the described changes - many of these could benefit from quantification. I am also missing experiments on the functional significance of this for mitotic exit.

Major points

The non-phosphorylatable mutant includes 3 alanine residues around the LxxIxE binding motif, but it is well established that the SP in the middle of PP2A-B56 binding sites increase binding strength (see Hertz et al, Mol Cell 2016). The only other phosphorylations likely to increase binding strength are immediately after the motifs, but there is none in that region within APC1. Therefore, ideally the mutant should be repeated with a single point mutant, which I assume will recapitulate the effect of the 3A.

Ideally the purified MBP-APC1 loop should be incubated with Cdk1 to see if this can increase binding strength. Peptides (-/+ phospho) could also be used to show binding is phospho-regulated. These first two points would clarify much better how Cdk1 primes Apc-B56 interaction.

A number of the changes in key figures are small and would benefit from quantification.

- I cannot see lower activity in fig2a. This should be quantified from multiple repeats. Same for EV2
- I cannot see a difference in activity (in B) whereas CDC20 binding is clearly different in A. The CycB degradation also looks only very marginally slower in anaphase extract in EV3
- In 3A, the B56-binding mutation are shown not to affect CDC20 binding (lanes 9 and 10) but in 3B they are suggested to affect binding (9 and 10). Again, these are small effects and should be quantified from multiple repeats.
- The pulldowns in 1C/D suffer from a lot of background binding which make the enrichment in WT less convincing. Is the MBP tag the issue and are these increases reproducible?

APC1 is dephosphorylated very rapidly (based on band shift) in the presence of PP2A, but the pS314/S318 signal is not. Has this dual phospho-antibody been previously been validated. If not, it is important to include this data, since this is a key result. Can the downward band shift be explained with pThr motifs being removed from APC1 (and if so which residue are these?).

It is not really possible to implicate Ser/Thr as the reason for the differential behaviour without

switching the Ser-Thr residue in APC1/Cdc20 to see if this produces different behaviour. Otherwise, it could simply be that the region on APC1 is protected from dephosphorylation (due to conformational constraints for example). The Cdc20 switch experiment has already been performed (Hein et al, NCB, 2017) although this paper is not cited in this context. Similar switching of Serines to Threonine on APC1 would be nice.

I think a big important missing part of the paper concerns the functional significance of this regulation by PP2A on mitotic exit. In particular, does it regulate SAC silencing/MCC disassembly or if not then what is the function for anaphase APC/C? PP2A-B56 activity is needed to silence the SAC (missing refs here) and so is PP1-Knl1. In fact, PP1-Knl1 also regulates CDC20 dephosphorylation and APC/C binding (paper by Desai group that is referenced). Considering the overlap with the Desai paper, this is not discussed enough. The PP1-Knl1 pathway works at kinetochores where PP2A-B56 is also localised - do the authors suspect that PP2A-B56 work in this region or generally in the cytoplasm? If it is the later then why do you need both pathway or are they working at slightly different times.

Minor points

Why would Cdk1 inhibit Cdc20 and then prime APC-B56 binding to remove this phosphorylation. The regulatory logic for this is not clear and couple perhaps be discussed

The B56 binding domain in highlight in 1A looks to be incorrect. Should this not be the LLSPVPE sequence further upstream?

1st Revision - authors' response

26 September 2019

RE: Manuscript EMBOR-2019-48503V1
Point-by-Point response to reviewers' comments

Referee #1:

*The presented data are of great importance for the cell cycle community because it addresses an important point, i.e. how is the activation of Cdc20 mediated under conditions when Cdk1 activity levels are still high. **This reviewer therefore thinks that the submitted ms should be published.** However, this reviewer is not always convinced by the presented data and the conclusions drawn by the authors. The authors should address the following points:*

In general, as outlined below the observed effects are very mild and sometimes even difficult to see. Therefore, to support the conclusions it is important that the authors quantify all experiments and show statistics from several experimentally independent experiments.

1. Fig. 1C: Specific binding over background binding of MBP-APC1-515-584 to WT APC1 is rather weak. The same applies for Fig. 3.

As requested, we have repeated the experiments three times and quantified all experiments. WT Apc1-loop⁵⁰⁰ does significantly bind to B56 in anaphase extract, compared with the same fragments with Δ11 or 3A mutation. The new data are presented as new Fig 1C with error bars.

2. The same applies to EV2B and EV2C: This reviewer has a hard time to see a difference in ubiquitylation efficiency of cyclin B and binding of Cdc20 to WT vs. Delta11 APC1.

As requested, we have repeated the experiments three times and quantified all experiments. The data are presented in the new EV2C and EV2D with error bars. WT apo-APC/C has higher cyclin ubiquitylating activity than the mutant APC/C harbouring Apc1-loop⁵⁰⁰ Δ11 (1-Δ11) (Fig EV2C). Consistently, WT Apc-APC/C binds more Cdc20 than mutant apo-APC/C harbouring 1-Δ11 does (Fig EV2D).

3. The same applies to Fig. EV3: The effects on the degradation of cyclin B are marginal.

We have repeated the experiments three times and quantified all experiments. The data are presented in the new Fig EV3.

4. Fig4A: It seems that the Cdc20-5A mutant still binds stronger to the wt than the mutant APC/C. In Fig EV5, the Cdc20-5A mutant seems to bind equally to both wt and mutant APC/C. Could the authors comment on this and provide quantifications?

We have repeated the experiments three times and quantified. The data are presented in the new Fig. 4C (and 4D) and EV5. In support of our model, unlike WT Cdc20, Cdc20-5A mutant seems to bind almost equally to both WT and mutant APC/C.

5. There is a strong effect of WT APC and the B56 binding deficient mutant on APC3 phosphorylation levels, e.g. EV2C. How the authors explain this. Please, comment.

We also think that the decreased Apc3 phosphorylation levels in B56-binding deficient mutants are interesting. As we mention in the text, the B56-binding site may have as-yet unidentified function in the regulation of APC/C activation directly or indirectly. At the moment, we do not know the mechanism(s).

6. Figure 4B: the authors know the exact binding site of B56 on the APC/C, so the binding mutant would be a good control in the in vitro phosphatase assay.

Our preliminary data suggest that this B56 binding site is not the only B56 binding site on the APC/C, so consequently detailed experiments uncovering all the B56 binding sites and their impacts are beyond the scope of this paper.

Minor Points:

- Page 3: the authors state that "there can be no APC/C activation without Cdk1..." and later that "...Cdh1 can activate both phosphorylated and unphosphorylated APC/C." These statements seem to be contradictory.

As requested, it has been corrected.

- Page 4: the authors use the term "short-conserved". Do they mean "short and conserved"?

As requested, it has been corrected.

- Page 4: the authors write that Cdk1 activity is high in anaphase. Is this specific for the way the *Xenopus* extracts used here are prepared? In normal mitotic cell divisions, Cdk1 activity is decreasing substantially in (early) anaphase.

Thank you for asking this. We have changed to "in anaphase extracts induced by non-degradable cyclin B". As such, Cdk1 activity is maintained at a high level without cyclin destruction.

- Page 4: the part where the authors talk about the activities of PP2A enzymes in cell cycle. It is written that PP2A-B55 is vital for mitotic exit and later that it is inactive in mitosis, which is somewhat contradictory/imprecise. In addition, the Nilsson lab has shown that B55-PP2A is important for Cdc20 dephosphorylation at mitotic exit once it is activated. Maybe this should be mentioned here as well.

As requested, it has been corrected and the description added.

- Figure 1A: In my printout the lines with the sequences are shifted so that corresponding amino acids are not in a column and the green box is not on the motif.

As requested, it has been corrected.

- Page 6 and Figure 1C: the authors show that the 3A mutant is compromised in B56 recruitment and suggest that this due to the absence of phosphorylation (by Cdk1). Is there more evidence that these sites (especially the serine in the B56-binding motif) are indeed phosphorylated (by Cdk1) in this context?

We have performed an in vitro Cdk phosphorylation assay and indeed WT Apc1-loop⁵⁰⁰ was well-phosphorylated. Conversely, the same fragment with 3A mutations was hardly phosphorylated (new

Fig 1D). Furthermore, purified B56 was loaded onto Apc1-loop⁵⁰⁰, in a Cdk phosphorylation manner. The results are presented in new Fig 1E and F.

- Page 10: it is unclear why a Cdc20 with 5 alanine mutations was used when before it was mentioned that only 3 threonine residues are important. I also couldn't find the information about the identity of the mutated amino acids

Cdc20 N-terminus (N-Cdc20) has five conserved Cdk sites. Phosphorylation of N-Cdc20 has been reported to be inhibitory to Cdc20-dependent APC/C activation (Labit, et al., EMBO J., 2012). Cdc20-5A has all the Cdk sites with alanine substitutions (S50A, T64A, T68A, T79A and S114A) and 5A is a well characterised mutant Cdc20 which is free from Cdk-mediated inhibition and has a higher affinity to the APC/C in anaphase (Labit et al., EMBO J., 2012). It has also been shown that phosphorylation of three threonine residues is sufficient (importantly T79 in particular) to inhibit Cdc20 function, but Cdc20-5A has been more commonly used as a Cdc20 mutant free from Cdk1-mediated inhibition.

The reference for Cdc20-5A has been added in the text.

Referee #2:

Overall this is a nice focused paper on a topic that could have important implications. On the downside, I was not overly convinced with some of the described changes - many of these could benefit from quantification. I am also missing experiments on the functional significance of this for mitotic exit.

Major points

The non-phosphorylatable mutant includes 3 alanine residues around the LxxIxE binding motif, but it is well established that the SP in the middle of PP2A-B56 binding sites increase binding strength (see Hertz et al, Mol Cell 2016). The only other phosphorylations likely to increase binding strength are immediately after the motifs, but there is none in that region within APC1. Therefore, ideally the mutant should be repeated with a single point mutant, which I assume will recapitulate the effect of the 3A.

As requested, we have made a single point mutant S558A within the B56 binding motif and investigated B56 binding activity. In support of our model, this S558A mutation reduced B56 binding. The new data are presented as new Fig. EV1B.

Ideally the purified MBP-APC1 loop should be incubated with Cdk1 to see if this can increase binding strength. Peptides (-/+ 6phosphor) could also be used to show binding is 6phosphor-regulated. These first two points would clarify much better how Cdk1 primes Apc-B56 interaction.

As suggested, we have investigated whether Cdk phosphorylation of Apc1-loop⁵⁰⁰ stimulates B56 binding. Our new result clearly indicates that upon phosphorylation WT Apc1-loop⁵⁰⁰ binds more efficiently to PP2A B56 than the Cdk site mutant (3A). The new data are presented as new Fig. 1E and F, highlighting that Cdk phosphorylation can prime Apc1-B56 interaction.

*A number of the changes in key figures are small and would benefit from quantification.
- I cannot see lower activity in fig2a. This should be quantified from multiple repeats. Same for EV2
- I cannot see a difference in activity (in B) whereas CDC20 binding is clearly different in A. The CycB degradation also looks only very marginally slower in anaphase extract in EV3.*

As requested, we have repeated the experiments three times and quantified. The new data clearly support our model. The previous Fig 2A, Fig EV2B and EV2C are replaced by new data and presented as new Fig 2A, Fig EV2C and EV2D, respectively. The quantification results are also presented.

Please note that Cdc20-binding data in the previous EV2A (new EV2B) are relevant to previous Fig. 2B (new Fig 2C), not EV2B (new EV2C). Similarly, Cdc20 binding data (previous EV2C; new EV2D) are relevant to previous EV2B (new EV2C).

- In 3A, the B56-binding mutation are shown not to affect CDC20 binding (lanes 9 and 10) but in 3B they are suggested to affect binding (9 and 10). Again, these are small effects and should be quantified from multiple repeats.

As requested, we have repeated the experiments three times and quantified. In interphase (Fig. 3A), the B56-binding mutations had little effect on Cdc20 binding whereas in anaphase (Fig. 3B) they had significant impact, resulting in reduction of Cdc20 binding. The data are presented in the new Fig 3A and B with quantification.

- The pulldowns in 1C/D suffer from a lot of background binding which make the enrichment in WT less convincing. Is the MBP tag the issue and are these increases reproducible?

As requested, we have repeated the experiments three times and quantified all experiments. WT Apc1-loop⁵⁰⁰ does significantly bind to B56 in anaphase extract, compared with the same fragments with Δ 11 or 3A mutation. The new data are presented as new Fig. 1C with quantification. Specific binding of Apc1-loop⁵⁰⁰ to B56g is also reproducible. The new data are presented as new Fig. EV1C with quantification.

APC1 is dephosphorylated very rapidly (based on band shift) in the presence of PP2A, but the pS314/S318 signal is not. Has this dual phospho-antibody been previously been validated. If not, it is important to include this data, since this is a key result. Can the downward band shift be explained with pThr motifs being removed from APC1 (and if so which residue are these?).

The phospho-specific antibodies, anti-pS314/S318 have been used before and published in Fujimitsu et al., Science 2016 (see Fig S8). So, the reference has been added in the text.

In terms of the downward band shift of Apc1 on the immunoblot using anti-pS314/pS318 antibodies, as the reviewer also speculates, other pThr sites somewhere in Apc1 are likely to be dephosphorylated. At the moment we don't know the exact sites. Apc1 has 110 Thr sites and seven of these are TP sites. We believe that the detailed analysis of pThr sites responsible for the downward band shift is beyond the scope of this paper.

It is not really possible to implicate Ser/Thr as the reason for the differential behaviour without switching the Ser-Thr residue in APC1/Cdc20 to see if this produces different behaviour. Otherwise, it could simply be that the region on APC1 is protected from dephosphorylation (due to conformational constraints for example). The Cdc20 switch experiment has already been performed (Hein et al, NCB, 2017) although this paper is not cited in this context. Similar switching of Serines to Threonine on APC1 would be nice.

We have already shown that *Xenopus* Cdc20 N-terminal fragment containing TP sites is rapidly dephosphorylated in anaphase extracts (See Fig S7; Labit et al., EMBO, 2012). Consistently, the Nilsson lab has also shown that human Cdc20 N-terminal fragment containing TP sites is rapidly dephosphorylated whereas the same fragment with SP sites is hardly dephosphorylated (Hein et al., NCB, 2017). We have now performed the Apc1-loop³⁰⁰ switching experiment. ³²P-labelled WT Apc1-loop³⁰⁰ fragment was poorly dephosphorylated in anaphase extracts whereas the same fragment with TP sites (7T) was very rapidly dephosphorylated under the same conditions. The new data support our model and suggest that the Ser/Thr specificity on PP2A active in mitosis plays a role for Cdc20-APC/C activation. The new data are presented as new Fig 4A and B with quantification.

Intriguingly, ³²P-labelled WT Apc1-loop³⁰⁰ was further phosphorylated in anaphase extracts, resulting in an upper band shift during incubation (Fig 4A, lanes 1-4). We assume that mitotic kinases other than Cdk (e.g. polo kinase) could phosphorylate WT Apc1-loop³⁰⁰ and induce band shifts, but we don't know the identity of the kinase(s) at the moment.

I think a big important missing part of the paper concerns the functional significance of this regulation by PP2A on mitotic exit. In particular, does it regulate SAC silencing/MCC disassembly or if not then what is the function for anaphase APC/C? PP2A-B56 activity is needed to silence the SAC (missing refs here) and so is PP1-Knl1. In fact, PP1-Knl1 also regulates CDC20 dephosphorylation and APC/C binding (paper by Desai group that is referenced). Considering the overlap with the Desai paper, this is not discussed enough. The PP1-Knl1 pathway works at kinetochores where PP2A-B56 is also localised - do the authors suspect that PP2A-B56 work in this region or generally in the cytoplasm? If it is the later then why do you need both pathway or are they working at slightly different times.

At present, we don't know whether and how PP2A recruited on Apc1-loop⁵⁰⁰ regulates SAC silencing/MCC disassembly. Early embryos or oocytes of most metazoans lack SAC function. Thus, our study demonstrates that Apc1-loop⁵⁰⁰ promotes APC/C-Cdc20 complex formation under normal conditions and is not dependent upon the SAC as it is not active in *Xenopus* egg extracts. In addition, our data strongly suggest that Apc1-loop⁵⁰⁰ promotes APC/C-Cdc20 complex formation through PP2A-B56-mediated Cdc20 dephosphorylation.

As the reviewer suggested, the relationship of PP2A-B56 on Apc1-loop⁵⁰⁰ and PP1-Knl1 is an interesting point. During this revision, a very interesting paper has been come out from the Saurin lab, showing that PP1 and PP2A-B56 have opposite phospho-dependencies. Phosphorylation has an inhibitory impact on PP1-SLiM recognition but enhances PP2A-B56-SLiM recognition. Consistent with this report, B56-binding to Apc1-loop⁵⁰⁰ is dependent on phosphorylation. The PP2A-B56 recruited on Apc1-loop⁵⁰⁰ may be more important in a situation or specific area where a mitotic kinase is high and PP1 is less responsive. A brief discussion has been added to the text.

We believe that detailed analysis of the relationship between PP2A-B56-Apc1-loop⁵⁰⁰ and PP1-Knl1 in APC/C regulation is beyond the scope of this paper. In this work, we investigated a novel B56-binding site within the APC/C for the first time and showed that B56 recruitment promotes APC/C-Cdc20 complex formation through dephosphorylation of Cdc20. Thus, we think that our work reveals a mechanism explaining how a mitotic co-activator Cdc20 can be dephosphorylated at the correct time, allowing its binding and subsequent activation of the APC/C, a vital ubiquitin ligase controlling the cell division.

Minor points

Why would Cdk1 inhibit Cdc20 and then prime APC-B56 binding to remove this phosphorylation. The regulatory logic for this is not clear and couple perhaps be discussed

Cdk1-dependent phosphorylation of N-Cdc20, in particular three threonine residues near the C-box (T64, T68 and T79 in *Xenopus*) inhibits Cdc20 loading to the APC/C (Labit et al., EMBO J., 2012). As phosphorylation of the APC/C has a positive impact on Cdc20-APC/C complex formation, Cdc20 phosphorylation should have an inhibitory impact, otherwise it is prematurely activated or uncontrolled. Yet, the APC/C needs to be activated in mitosis with a certain time lag. In order to achieve high APC/C-Cdc20 activity at the right moment and with a timed duration peak (i.e. no premature activation, rapid activation upon Cdk1 inactivation), it makes sense that Cdk1 inhibits Cdc20 and then primes APC/C-B56 binding to achieve de-phosphorylation.

This idea is also in agreement with a mathematical modelling for cell cycle oscillations during *Xenopus* early development. Cdk activates the APC/C and simultaneously antagonises Cdc20, which allows cycling as long as Cdc20 is phosphorylated and dephosphorylated faster than the APC/C (Ciliberto et al., Cell Cycle, 2005). As we demonstrate here, Apc1-loop³⁰⁰ is indeed more slowly dephosphorylated than phospho-threonine residues in N-Cdc20 in anaphase. Phosphatases, such as PP2A-B56, that specifically target Cdc20 even while CDK is active, would ensure this. Once the APC/C becomes active, degradation of cyclin B lowers CDK activity, which in turn decreases phosphorylation of Cdc20 and the positive feedback loop allows degradation of numerous APC/C substrates and exit from mitosis.

Though we are still far from a full understanding of the underlying mechanisms, this idea partly explains the regulatory logic for this.

The B56 binding domain in highlight in 1A looks to be incorrect. Should this not be the LLSPVPE sequence further upstream?

Thank you very much for spotting this error. It seems that misalignment conversion has happened during uploading our figure. The reviewer is correct; the highlighted sequence should be LLSPVPE. We have now corrected accordingly.

2nd Editorial Decision

28 October 2019

Thank you for submitting the revised version of your manuscript. It has now been seen by both of the original referees.

As you can see, the referees find that the study is significantly improved during revision and recommend publication. Before I can accept the manuscript, I need you to address some minor points below:

- Please provide 3-5 keywords for your study. These will be visible in the html version of the paper and on PubMed and will help increase the discoverability of your work.
- A 'table of contents' should be added to "Appendix Figure S1" to indicate that the appendix only contains this figure.
- Papers published in EMBO Reports include a 'Synopsis' to further enhance discoverability. Synopses are displayed on the html version of the paper and are freely accessible to all readers. The synopsis includes a short standfirst summarizing the study in 1 or 2 sentences that summarize the key findings of the paper and are provided by the authors and streamlined by the handling editor. I would therefore ask you to include your synopsis blurb.
- In addition, please provide an image for the synopsis. This image should provide a rapid overview of the question addressed in the study but still needs to be kept fairly modest since the image size cannot exceed 550x400 pixels. For example, the synopsis image can be adapted from figure 4F.
- Our production/data editors have asked you to clarify several points in the figure legends (see attached document). Please incorporate these changes in the attached word document and return it with track changes activated.

Thank you again for giving us to consider your manuscript for EMBO Reports, I look forward to your minor revision.

REFeree REPORTS

Referee #1:

The authors have spent a lot of work to address my comments and concerns. The resulting data strongly support the main conclusion of the manuscript. I therefore recommend publication of the ms as it is.

Referee #2:

The authors have adequately addressed all my concerns and I think the manuscript has now improved as a result of the changes.

2nd Revision - authors' response

31 October 2019

1. Referee 1, points 1-4

As suggested by the referee, we repeated experiments and quantified all the experiments. The new data are presented with error bars (new Fig 1C, Fig EV2C, Fig EV2D, Fig EV3, Fig 4C, 4D and Fig EV5). Importantly, all the new results are consistent with our conclusions, strengthening our manuscript.

2. Referee 1, asking evidence of phosphorylation of Apc1-loop500 by Cdk

We have performed an in vitro Cdk phosphorylation assay and confirmed that WT Apc1-loop500 is wellphosphorylated by Cdk. The new data are presented in new Fig 1D. Furthermore, we have reconstituted Cdk phosphorylation-dependent B56 loading onto Apc1-loop500. The new results are presented in new Fig 1E and F. These data strongly support our conclusions and model (Fig. 4F).

3. Referee 2, point 1

As requested, we have made a single point mutant S558A within the B56 binding motif and investigated B56 binding activity. In support of our model, S558A mutation reduced B56 binding. The new data are presented as new Fig. EV1B.

4. Referee 2, point 2; how Cdk primes APC/C-B56 interaction

As mentioned above, in referee 1 point 2, we have investigated whether Cdk phosphorylation of Apc1-loop500 stimulates B56 binding. Our new result clearly indicates that upon phosphorylation Apc1-loop500 binds more efficiently to PP2A B56 than the Cdk site mutant (3A). The new data are presented as new Fig. 1E and F, highlighting that Cdk phosphorylation can prime Apc1-B56 interaction.

5. Referee 2, asking quantification of the data

This is similar to above referee 1 point 1. We repeated experiments and quantified all the experiments as requested. The new data are presented as new Fig EV2C, EV2D, Fig 3A, 3B, Fig 1C, EV1C, clearly supporting our conclusions.

6. Referee 2, asking experiment with switching of serine to threonine on Apc1

It took some time to make the Apc1 construct in which all seven serine residues are changed to threonine and prepare the protein (7T) for experiments. But as it would strengthen our model if successful, we have done it. ³²P-labeled WT Apc1-loop300 fragment was poorly dephosphorylated in anaphase extracts whereas the same fragment with TP sites (7T) was very rapidly dephosphorylated. The new data support our model and are presented as new Fig 4A and B with quantification.

3rd Editorial Decision

11 November 2019

Thank you for submitting your revised manuscript. I have now taken a careful look at everything and all looks fine. Therefore I am very pleased to accept your manuscript for publication in EMBO Reports.

Corresponding Author Name: John E. Burke

Journal Submitted to: EMBO Rep

Manuscript Number: EMBOR-2019-48441V1